# Radiological Predictors of Cognitive Impairment in Paediatric Brain Tumours Using Multiparametric Magnetic Resonance Imaging: A Review of Current Practice, Challenges and Future Directions

**DOI:** 10.3390/cancers17060947

**Published:** 2025-03-11

**Authors:** Simon Dockrell, Martin G. McCabe, Ian Kamaly-Asl, John-Paul Kilday, Stavros M. Stivaros

**Affiliations:** 1Division of Informatics, Imaging & Data Sciences, University of Manchester, Manchester M13 9PL, UK; simon.dockrell@nhs.net; 2Children’s Brain Tumour Research Network, Royal Manchester Children’s Hospital, Manchester University NHS Foundation Trust, Manchester M13 9WL, UK; ian.kamaly@nhs.net (I.K.-A.); john-paul.kilday@mft.nhs.uk (J.-P.K.); 3The Geoffrey Jefferson Brain Research Centre, Northern Care Alliance NHS Foundation Trust, Salford M6 8FJ, UK; martin.mccabe@manchester.ac.uk; 4The Christie NHS Foundation Trust, Manchester M0 4BX, UK; 5Division of Cancer Sciences, University of Manchester, Manchester M13 9PL, UK

**Keywords:** artificial intelligence, brain tumours, central nervous system (CNS), children, magnetic resonance imaging (MRI), cognition, cognitive outcomes, late effects, machine learning, paediatrics

## Abstract

While many children with brain tumours survive to adulthood, brain damage from tumours and their treatment impact future quality of life. Cognitive impairment, which involves thinking processes, learning and academic performance, is a key issue. Many factors contribute to cognitive impairment with tumours occurring at any age, in different locations with various tumour types requiring different treatments. Patients have varying combinations of risk factors, making it difficult for traditional statistical techniques to determine the causes of cognitive impairment. This review focuses on how brain imaging can be used to predict cognitive impairment. We discuss the challenges and possible solutions for this research including the need for large patient numbers requiring multi-site collaboration and variations in imaging performed. We discuss how imaging data can be combined with health and treatment data using artificial intelligence techniques to identify the key drivers of cognitive impairment and those children likely to be at high-risk.

## 1. Introduction

With modern treatment, the majority of children with brain tumours become long-term survivors [1]. However, damage from brain tumours and their treatment can be detrimental to cognition and future quality of life [2]. Lower processing speed, attention, working memory, visual memory, visuospatial processing, and executive function scores are reported among adult survivors, which correlate with lower educational achievement [3]. They are more likely to be unemployed, on lower incomes and have psychological morbidity, while less likely to graduate university or sustain long-term relationships [3,4]. Improving understanding of the causes of cognitive impairment may permit changes in treatment, reducing future cognitive decline. Multiple interventions show promise in improving long-term cognitive outcomes including cognitive rehabilitation [5], pharmacological interventions [6] and lifestyle changes [7]. Therefore, early identification of high-risk children, enabling intensive rehabilitation and support, may benefit long-term outcomes.

While the causes of cognitive impairment in children with brain tumours are complex and multifactorial [8], many can be observed on brain imaging. Magnetic Resonance Imaging (MRI) can demonstrate tumour location, size, and secondary conditions, such as hydrocephalus. Advanced MRI sequences can demonstrate physiological processes occurring within tumours and the brain [9,10]. Longitudinal imaging provides insights into changes occurring over time. Multiparametric MRI scans are increasingly used for research and outcome prediction in adult and paediatric neuro-oncology, such as predicting tumour grade [11,12], tumour histology type [12,13,14,15], molecular subtype [16] and prognosis including survival, risk of metastases and recurrence [17,18]. Challenges to collaborative research in paediatric neuro-oncology include small patient numbers for tumour subtypes [19], the use of historical imaging and variability in scan acquisition [9,20]. In the future, larger collaborative studies standardising imaging parameters and harmonising scans are needed. Large, shared data repositories from multiple institutions may provide the patient numbers required for artificial intelligence techniques, which can combine imaging, clinical and demographic data for outcome prediction [21].

## 2. Materials and Methods

An electronic literature search was performed using the PubMed database by a single reviewer. Relevant studies published prior to November 2024 were included with no limitations on year of publication. The search strategy focused on title or abstract keywords in the following categories: paediatric (including “paediatric” or “pediatric” or “children”), brain tumours (including “brain tumours” or “neuro-oncology”), radiomics (including “radiomics” or “radiogenomics”), MRI (including “MRI” or “Magnetic Resonance Imaging”), cognitive (including “cognitive” or “cognition” or “neurocognitive” or “neuropsychological”), outcomes (including “outcomes” or “late-effects”) and artificial intelligence (including “artificial intelligence” or “AI” or “machine learning” or “deep learning”). Reference lists from relevant papers were also reviewed. This was a broad narrative review, so there were no specific inclusion or exclusion criteria. The results of the initial search identified 257 papers of interest. This was subsequently reduced to 150 papers within the final manuscript, following consensus evaluation by all authors.

## 3. Imaging Modalities and Their Uses in Paediatric Neuro-Oncology

### 3.1. Conventional MRI

MRI scans are the most common imaging modality used for investigation, treatment-planning and surveillance of brain tumours. Different MRI sequence acquisitions can display many characteristics of tumours and the surrounding brain as well as help distinguish tumours from other differential diagnoses (Figure 1 and Figure 2). Radiological features including tumour size, location, enhancement patterns and heterogeneity as well as the presence of cysts, necrosis, haemorrhage, and calcification aid radiologists’ prediction of tumour type (Table 1) [22].

### 3.2. Advanced MRI

The introduction of advanced parametric imaging techniques has enabled improved understanding of brain and tumour physiology using diffusion, perfusion and spectroscopic characteristics. These enhance radiologists’ ability to predict tumour grade, histological type and molecular subtype as well as identify recurrence and contribute to models predicting prognosis (Table 2).

Diffusion-weighted imaging (DWI) sequences display the random movement of water molecules within the brain and cerebrospinal fluid (CSF) [10,23,24]. As DWI sequences are influenced by T2 as well as diffusion properties of tissues [23], an apparent diffusion coefficient (ADC) map is calculated with the signal intensity quantifying diffusion within each voxel. High tumour cellularity, which is more common in high-grade tumours, restricts diffusion [14] and produces low ADC values compared to the surrounding brain (Figure 3), permitting differentiation between tumour types [24]. DWI can identify tumour infiltration within areas of vasogenic oedema and guide biopsy targets in heterogenous tumours [10]. As myelinated white matter (WM) restricts diffusion, DWI can be used to monitor WM development and post-treatment demyelination. DWI can identify post-operative injury and cytotoxic oedema caused by influx of intracellular water molecules when cell metabolism fails [10]. In DWI, acute ischaemia restricts diffusion, while established infarction causes necrosis, reducing cellularity and increasing diffusion [27].

Diffusion tensor imaging (DTI) quantifies the amount and direction of diffusion in multiple directions (Figure 4). Mean diffusivity (MD) measures the mean amount of total diffusion in a voxel, while fractional anisotropy (FA) measures how random the diffusion direction is [28]. Isotropic diffusion is random and therefore equal in all directions, whereas impediments to diffusion can limit diffusion in certain directions, producing anisotropic diffusion. WM tracts are linear and densely packed, restricting perpendicular diffusion and causing directional diffusion parallel to the WM fibres, which produces low MD and high FA. Diffusion parallel to WM fibres can be measured with axial diffusivity (AD), while diffusion perpendicular to WM fibres can be measured with radial diffusivity (RD) [29]. Measuring diffusion in multiple directions enables estimation of the primary diffusion direction and orientation of fibres within WM tracts [29]. However, conventional deterministic tractography techniques only display the mean diffusion direction within each voxel so struggle where fibres cross. More advanced probabilistic models can account for crossing fibres [24], but they are user-dependent as they are traced from a selected starting point [30]. Tumours and their treatments can disrupt tracts and cause WM atrophy, reducing diffusion restriction perpendicular to fibres so fibre integrity can be inferred from FA, MD, AD and RD [29,31,32].

Perfusion MRI sequences consist of multiple techniques, which quantify cerebral blood volume (CBV) or cerebral blood flow (CBF) to tumours and the surrounding brain. Tumour vasculature often has a deficient blood–brain barrier, increasing vascular permeability [24]. Dynamic Contrast-Enhanced (DCE) imaging is a T1-weighted sequence, which measures the diffusion of contrast into interstitial tissues, quantifying capillary permeability [10,24] where increased capillary permeability can correlate with higher-grade tumours [10]. Dynamic Susceptibility Contrast (DSC) are T2 or T2* sequences, which use contrast to measure loss of susceptibility-induced signal over time. They provide an estimate of relative cerebral blood volume (rCBV). rCBV quantifies blood vessel density and thus angiogenesis within a region of interest (ROI), which can help differentiate tumour types and grades [10,12]. DSC can highlight increased microvessel density within non-enhancing tumours and infiltrative oedema [10,24]. Arterial Spin Labelling (ASL) is a non-contrast technique using an inversion pulse to label blood as it enters, allowing it to be subtracted from the final image. It has swift acquisition times, little post-processing and images the whole brain (Figure 5). It provides an estimate of CBF, which can be measured within the tumour and compared to the patient’s normal grey and white matter. CBF has also shown benefits in differentiating tumour types [11] and grades [10].

The most common form of Magnetic Resonance Spectroscopy (MRS) uses hydrogen protons referred to as ^1^H MRS to analyse the signal of hydrogen protons linked to other molecules [33,34]. Following acquisition of an MRI scan, ^1^H MRS can be performed within single or multiple voxels, thereby producing a spectrum with different chemical metabolites producing peaks at characteristic locations on the spectrum (Figure 6) [23]. The area under generated peaks demonstrates the metabolite concentrations [33], which can be reported as absolute values or ratios to another reference metabolite peak. Different metabolite concentrations can highlight physiological processes occurring within the voxel [34]. MRS has shown applications in differentiating tumour types, detecting malignant transformation and distinguishing tumour recurrence from radionecrosis [33,34]. Clinical use can be limited by long acquisition times, challenges in small tumours and blood or calcification causing heterogeneity in the magnetic field [10,12,13,24].

## 4. Applications of Advanced Imaging Analysis for Prediction in Paediatric Neuro-Oncology

During clinical practice, radiologists routinely assess qualitative information and some semi-quantitative data, such as two-dimensional measurements [35,36]. However, there are vast amounts of untapped research data about children with brain tumours in their multiparametric MRI scans, and the potential for longitudinal monitoring via serial imaging. Radiomics is the term used for quantitative imaging data extracted from scans, which are used to guide clinical decision-making, research and predictive modelling [9,21,35,37]. Computer-aided quantitative analysis can include semantic data using automated techniques to extract information that radiologists can assess, such as comparing signal intensity within tumour to the surrounding brain, tumour shape, size and location [35,36]. However, they can also extract agnostic data which cannot be consciously seen, such as statistical analyses of voxel signal intensities [22,35,36]. Histogram analyses explore the range and frequency of signal intensities within images [14]. Textural analyses involve quantifying spatial patterns of voxel signal intensity distribution [13]. Table 3 outlines the different applications multiparametric MRI has been used for in paediatric neuro-oncology research.

Hara et al. predicted tumour type, neuroaxis metastases and recurrence from radiological phenotyping of 34 paediatric embryonal brain tumours, using post-contrast T1 and FLAIR [22]. They also identified radiomic features predictive of age, suggesting distinct phenotypes between younger and older children. Diffusion and perfusion imaging provide further information regarding the tumour microenvironment [11,12,13]. Rodriguez-Gutierrez et al. reviewed 40 children with infratentorial tumours using post-contrast T1, T2 and ADC biomarkers regarding shape, textural features, and histogram analysis [13]. ADC was superior at differentiating tumours compared to T1 and T2. ADC was used to differentiate medulloblastoma, ependymoma and pilocytic astrocytoma and between medulloblastoma subgroups. Grist et al. combined T2, ADC and DSC biomarkers to predict tumour type and grade. Both regional and whole-brain ADC measures could distinguish between tumour types and high-grade from low-grade tumours, while whole-brain CBV metrics could also discriminate between some tumours [12]. Combining biomarkers enhanced predictive ability. Hales et al. also demonstrated that tumour ROI, mean and minimum ADC, and maximum tumour CBF could differentiate high-grade from low-grade tumours and between some tumour types [11].

Machine or deep learning techniques can select the most effective variables and assess the predictive accuracy of models for use in clinical practice [21]. Integrating radiomics with demographic and clinical data can further enhance clinical prediction models (Figure 7) [43,44,45]. Mahootiha et al. used a pre-trained deep learning tool to extract imaging features from pLGG using pre-operative T2 sequences and combining them with clinical data [18]. They created low-risk and high-risk groups for post-operative tumour recurrence, reporting that the addition of imaging features improved risk prediction by 13%. The tools to extract scan data and integrate demographic and clinical data to predict outcomes therefore do exist but need to be adapted and applied for cognitive outcome prediction in paediatric neuro-oncology (Figure 7).

## 5. Factors Influencing Cognitive Outcomes

### 5.1. Tumour Location

Early cognitive outcome research classified brain tumours into supratentorial and infratentorial tumours focusing on intelligence quotient (IQ) outcome measures without statistically significant differences [46,47]. However, despite normative IQ scores, many children have more subtle issues contributing to disability [48,49]. Studies focusing on neuropsychological subtests can identify different cognitive processes impacted by tumour location [48,49,50,51,52,53]. There is discordance regarding whether supratentorial or infratentorial tumours cause more profound cognitive deficits [54]. Some papers report inferior cognitive outcomes with supratentorial tumours [46,47,49,55], others infratentorial tumours [56], while some report no difference [44,54,57,58]. Some papers subdivide the supratentorium, comparing supratentorial hemispheric, supratentorial midline and infratentorial tumours. The worst outcomes were reported in supratentorial hemispheric tumours and best outcomes in supratentorial midline tumours [59,60,61]. Infratentorial tumours can be subdivided into cerebellar and brainstem or extra-cerebellar. However, sub-analyses within these regions [49,62] or investigation of specific anatomical structures [63] is rarely performed due to small patient numbers. Meta-analyses are challenging due to differing tumour types, variation in cognitive tests and location classifications used. deRuiter et al. performed a meta-analysis which included assessment of supratentorial versus infratentorial tumour location in 677 paediatric patients with various tumour types from 19 papers, limiting outcomes to IQ measurements from the WISC-III test with no statistically significant differences reported [58]. Ultimately, the utility of studies focusing on such large brain regions is limited, often failing to identify the likely cognitive deficits encountered by individuals.

Tumours in different locations impact different cognitive processes [62,64] and the brain exhibits redundancy with plasticity following insults [64,65,66]. Corti et al. demonstrated that tumour location influenced which cognitive domains drove academic skills without detectable deficits [54]. Verbal and visuospatial memory correlated with mathematical performance in children with supratentorial tumours, whereas only visuospatial memory correlated for children with infratentorial tumours. Resting-state functional MRI (fMRI) studies have demonstrated neural reorganisation with functional connectivity changing within the default mode network, memory and frontal–basal ganglia circuits among paediatric brain tumour patients [64,65,66]. However, sub-analyses of tumour location were not possible in most studies investigating the impact of tumour location on cognitive outcomes due to small patient numbers. An exception is Traunwieser et al., who analysed 316 German children with pLGG recruited to the European paediatric LGG study SIOP-LGG 2004 [62]. Within their patient cohort, they reviewed tumour laterality and subdivided supratentorial midline tumours into thalamus, visual pathways or other. Infratentorial tumours were split into caudal brainstem lesions and cerebellar tumours, which were subdivided into right, left or vermian. Supratentorial hemispheric tumours had worse scores than supratentorial midline and infratentorial tumours for crystallised intelligence, which relies on the ability to recall and use previously learned knowledge or skills. However, sub-analysis revealed that statistically significant differences were only present in left-sided supratentorial hemispheric tumours. Larger prospective studies are needed to investigate the contribution of smaller brain regions and specific structures to cognitive outcomes. Analysis of damage to neighbouring structures, hydrocephalus and raised intracranial pressure are also likely to be beneficial.

### 5.2. Eloquence Classifications

Eloquent brain consists of structures which, if damaged, are likely to cause neurological or cognitive deficits. There are multiple brain eloquence classification systems, which are outlined in Table 4. However, these mostly predict neurological deficits in adults and have rudimentary eloquence gradings [67] with brain eloquence dichotomised into eloquent or not within the Shinoda, Chang and Spetzler-Martin classifications [68,69,70].

There is a need for eloquence classifications focusing on cognitive outcomes in children incorporating recent advances relating to the neuroanatomical correlates of cognitive outcomes. They should also include damage to eloquent structures at presentation and following treatment as opposed to tumour location alone. These can be detected radiologically as compression of eloquent structures [49], vasogenic oedema on T2 FLAIR [10,23,49], and ischaemia, cytotoxic oedema and WM damage on DWI or DTI [10,24,73]. Long-term reduction in perfusion [25], fMRI activity [64,65] and atrophy of structures post-treatment can also correlate with cognitive impairment [32,74].

### 5.3. White Matter Tracts

Damage to WM tracts is a significant risk factor for cognitive impairment in paediatric brain tumour patients [28,31,32,43,55,73,75]. WM can be injured by the tumour, vascular insults [76], surgery and chemotherapy [32,55], but is particularly sensitive to radiotherapy [28,75,77,78,79]. Radiological measures of WM health and integrity include segmented WM volumes (often in comparison to grey matter and CSF volumes) [80,81], T2 or FLAIR WM hyperintensities [76] and diffusion measures including ADC, MD, FA, AD and RD [28,31,32,55,77].

Global WM volume loss has been associated with lower IQ [32,75,80,82], attention [75,80], processing speed [32] and academic performance [75]. Reduction in WM FA and increased MD have been associated with lower IQ [32,77], processing speed [32,55], sustained attention and working memory [55]. Damage to multiple WM tracts has been associated with cognitive impairment in paediatric brain tumour patients including the corticospinal tracts [79], superior longitudinal fasiculi [31], inferior fronto-occipital fasiculi [28,79], uncinate fasiculi [28], the cingulum [31,81], corpus callosum [28,73], frontocerebellar tracts [32,83], internal capsules, corona radiata, post-thalamic radiations, sagittal striatum, external capsules and cerebral peduncles [31,73]. Radiotherapy and chemotherapy can induce cerebral microbleeds and vascular injury causing chronic small-vessel disease and demyelination seen as WM hyperintensities on T2 and FLAIR sequences [76,84]. Increasing WM hyperintensity volume has been correlated with lower IQ, processing speed, memory, visuospatial ability [84] and executive function [76,84].

### 5.4. Tumour Size

Larger tumours cause more compression of neighbouring structures, oedema and raised intracranial pressure. Increasing tumour diameter has been associated with lower FSIQ, perceptual reasoning and processing speed [44]. Tonnig-Olsson et al. found that among children presenting with symptoms of raised intracranial pressure, the mean maximal tumour diameter was larger in boys than girls [85]. They attributed this to their larger cranial vault accommodating a larger tumour volume before decompensation occurred, leading to raised intracranial pressure. Increasing tumour size negatively correlated with verbal IQ. Law et al. also reported that larger tumour size was associated with worse working memory outcomes in infratentorial tumours [83], but diameter over 4.5 cm [86] or 5 cm [87,88] is a recognised risk factor for post-operative paediatric cerebellar mutism syndrome (pCMS), which is associated with long-term cognitive impairment [52,78,79]. In contrast, other papers found no correlation between tumour size and cognitive outcomes [48,49]. Patients with seizures may present earlier when tumours are smaller [89]. Therefore, epilepsy can be a confounding cause of cognitive impairment in patients with smaller tumours [49]. Measures of tumour size and distortion of brain structures, such as sulcal effacement, midline shift and tonsillar descent, can demonstrate raised intracranial pressure radiologically [49].

### 5.5. Radiological Phenotyping

High-grade tumours often require more intensive treatment than low-grade tumours, typically incorporating radiotherapy and chemotherapy, both of which are associated with cognitive impairment. However, children with high-grade tumours have shown increased risk of cognitive impairment compared to low-grade tumours prior to any treatment [53,90]. Radiological phenotyping based on tumours’ radiological characteristics on T1, T2, FLAIR, diffusion, perfusion, and MRS metrics show promise in differentiating histological groups of paediatric tumours from similar locations and differing grade within tumour entities [11,12,13,17,22,33]. The embryonal tumours medulloblastoma [91] and ATRT [92] are at increased risk of cognitive impairment compared to pLGG, which may relate to the craniospinal radiotherapy routinely used outside of infancy and a relatively high risk of pCMS compared to other infratentorial tumours [86,88,93]. Moreover, some differences in cognitive outcomes have been reported between certain molecular subgroups of medulloblastoma [94]. Multiple studies demonstrate distinct radiological phenotypes for different molecular subgroups of medulloblastoma [16], ependymoma [38], ATRT [39], DIPG [40] and pLGG [41]. Given the association between certain tumour subtypes and particular locations [74], radiological tumour phenotyping may help predict structures likely to be damaged and future treatments received. Consequently, radiological tumour phenotyping is likely to benefit cognitive outcome prediction.

### 5.6. Epilepsy

Seizures are more common with supratentorial tumours, particularly in the temporal or frontal lobes [95], and can occur at presentation or following treatment [96]. Epilepsy is common among certain low-grade tumours including dysembryoplastic neuroepithelial tumours, ganglioglioma, oligodendroglioma and astrocytomas [95,96]. In children with brain tumours, epilepsy and anti-epileptic medications can contribute to cognitive impairment [48,97]. Iuvone et al. found that increasing severity of epilepsy correlated with lower IQ scores [49]. The worst cognitive deficits were in patients with mesial temporal lobe tumours causing severe epilepsy. Treatment with an increasing number of anti-epileptic medications is correlated with lower non-verbal reasoning, working memory, processing speed and cognitive flexibility [98]. Early surgery for epileptogenic tumours permitting discontinuation of anti-epileptic medications is associated with better cognitive outcomes [99]. Irestorm et al. report that despite children presenting with seizures tending to have smaller tumours, after controlling for tumour size, seizures are associated with lower cognitive scores prior to surgery [89]. Tsai et al. reported that paediatric brain tumour patients presenting with seizures were more likely to be benign tumours with no neurological deficits, while patients with malignant tumours were less likely to present with seizures, but more likely to have focal neurological deficits and signs of raised intracranial pressure [95]. Tumours in epileptogenic locations or that can be radiologically phenotyped as epileptogenic tumours may influence cognitive outcome prediction.

### 5.7. Hydrocephalus

Children with brain tumours can develop obstructive hydrocephalus or communicating hydrocephalus from tumour dissemination, post-surgical changes or increased CSF viscosity and volume [100,101]. Patients can require CSF diversion at presentation, some may have hydrocephalus that resolves with treatment of the tumour, while others require long-term CSF diversion with ventriculoperitoneal shunt insertion or endoscopic third ventriculostomy (ETV). Many papers report that hydrocephalus is associated with lower cognitive test scores soon after presentation [53,62,89,102]. With treatment, test scores can improve without long-term cognitive deficits [48,59,90,101], but persistent ventriculomegaly is associated with cognitive impairment [48]. In contrast, other papers report that treatment for hydrocephalus is associated with long-term cognitive deficits [48,52,53,55,61,78,82,97,101]. Ventriculoperitoneal shunt malfunction, ETV failure or CSF infection require further surgeries, which are associated with cognitive impairment [103]. Ventricular size can be estimated radiologically using multiple techniques including the Evans index, cella media index, ventricular angle [102], fronto-occipital horn ratio [104], the bicaudate ratio [105], temporal horn or third ventricle measurements as well as segmented ventricular volume. However, ventricular size varies between people and there is usually no premorbid imaging for comparison. Other signs of hydrocephalus and raised intracranial pressure include transependymal CSF seepage, causing periventricular hyperintensity on FLAIR [89,101] and sulcal effacement [49,62]. Ventricular volumes dynamically change with treatment over time, so longitudinal assessment may aid the prediction of cognitive outcomes [104].

### 5.8. Treatment-Related Cognitive Impairment

#### 5.8.1. Surgery

Surgery for brain tumours can damage brain tissue during the approach to the tumour, causing vascular insults and retraction injury. Studies investigating children with brain tumours treated with surgery only report lower cognitive tests scores within 3 months and at long-term follow-up compared to healthy siblings [106] and the general population [107]. However, most reported studies lack pre-operative cognitive testing, so the impact of tumour acquired brain injury and surgery cannot be differentiated. Fraley et al. followed 11 patients with a mean pre-operative FSIQ of 105.3, which declined to 102 after 6 months, but recovered to 104.9 at 2 years post-surgery [90]. Stargatt et al. reported no statistically significant differences in IQ, processing speed and attention scores between 6 infratentorial tumour patients with pre-operative cognitive testing and 23 tested soon after surgery [82]. Weusthof et al. reported that in a cohort of 47 paediatric brain tumours in various locations treated with surgery only, 6% of children failed a school year, 11% had to change school and 23% reported subjective cognitive impairment [56]. However, these case series have very small patient numbers, limiting the conclusions which can be made.

Extent of resection is important, as gross total resection may damage more brain than partial resection or biopsy. Longitudinal follow-up is necessary as early deficits following gross-total resection may improve or be offset by subsequent tumour growth or adjuvant therapy in partial resection or biopsy groups [56]. However, tumour location differs between these groups, contributing to different surgical decision-making [59,62]. Multiple surgeries are associated with increased risk of cognitive impairment [55,103,108,109], and surgical complications are associated with an increased incidence of repeating a school year [110]. The extent of resection and post-operative complications, such as oedema, ischaemia and hydrocephalus can be assessed radiologically. Larger prospective studies are required with pre- and post-resection cognitive testing to improve understanding of the impacts of surgery on cognition.

#### 5.8.2. Post-Operative Paediatric Cerebellar Mutism Syndrome

pCMS involves delayed-onset mutism or dyspraxic speech usually occurring after resection of infratentorial tumours. It is thought to be caused by damage to the proximal efferent cerebellar pathway (pECP), also known as the dentato-thalamo-cortical tract (DTCT), between the cerebellum and brainstem during surgery [111]. While patients usually recover, this is often incomplete, and pCMS is associated with long-term cognitive deficits [52,78,79]. Proposed risk factors for pCMS include midline rostral fourth-ventricle tumour location [93], increasing tumour size [86,87,88], medulloblastoma or ATRT histology [88,93], younger age [93], and damage to the fastigial nuclei [112], the middle [113] or superior cerebellar peduncles, dentate nuclei, brainstem and cerebellar vermis [63,87,111]. Recent research has demonstrated no difference with vermis-sparing surgical approaches, suggesting the size of vermian split may simply reflect tumour size [93]. Radiological assessment of tumour location, size and damage to relevant structures neighbouring the fourth ventricles along with radiological phenotypes suggesting medulloblastoma or ATRT are likely to improve prediction of pCMS [113]. Hypertrophic olivarian deterioration on proton density or T2 MRI has been identified as a surrogate marker of pECP damage and predictor of pCMS [111,114]. Damage to the pECP on DTI [115] and widespread cortical decreased CBF on DSC particularly in the frontal regions have also been proposed as radiological markers of pCMS [116].

#### 5.8.3. Radiotherapy

Craniospinal irradiation is a significant risk factor for cognitive impairment [46,48,57,58,59,60,90]. Radiotherapy is detrimental to WM development, particularly in younger children [75], and is strongly associated with impairments of general and specific intelligence, which are more profound in younger children [47,57,60,75,85]. Higher radiotherapy doses are associated with increased WM volume loss and lower neuropsychological test scores [75,77]. Recent discoveries have highlighted that there is some genetic susceptibility to radiation-induced cognitive damage [117]. Reduced-dose radiotherapy regimes for low-risk tumours or combinations with chemotherapy reduce WM damage and cognitive impairment [118,119]. Technological advances of 3D conformal radiotherapy have enabled more targeted radiotherapy [108], while proton beam radiotherapy reduces the exit dose distal to the tumour, decreasing cognitive impairment [56,78]. These technological advances can make use of better understanding of radiosensitive eloquent structures, such as the hippocampus, to minimise the radiation dose they receive and reduce radiation-induced cognitive impairment [43,120]. Radiation-induced WM atrophy and loss of integrity can be seen on DTI [73], while radiation-induced cerebrovascular disease can be demonstrated by microbleeds on DWI or SWI, FLAIR WM hyperintensities and decreased CBF on perfusion sequences [25]. Radionecrosis is delayed tissue death, which can occur following radiotherapy. It can be identified as new areas of enhancement post-radiotherapy which resolve over time [121] or can be differentiated from tumour progression using MRS [33,34].

#### 5.8.4. Chemotherapy

Many chemotherapeutic agents are associated with cognitive impairment [122]. However, most studies investigating cognitive outcomes in paediatric brain tumours do not distinguish between the effects of radiotherapy and chemotherapy. Some historical papers report that the addition of chemotherapy to surgery and radiotherapy caused no further detriment [59,123]. Some papers report improved outcomes as chemotherapy potentially permitted less aggressive surgery [48] or radiotherapy [118]. Other papers report worse outcomes but failed to differentiate the effects of chemotherapy from the tumour at baseline and any subsequent surgery [62,108,109]. Treatment with intrathecal methotrexate is associated with WM damage [124] and worse cognitive outcomes in children under 10 years old [124]. While Neurofibromatosis Type 1 (NF-1) patients are predisposed to developing brain tumours, it is also associated with cognitive impairment without any brain tumours [125]. Chemotherapy is often the preferred treatment for NF-1-associated tumours, avoiding surgery and radiotherapy. However, NF-1 patients undergoing chemotherapy appear more susceptible to WM damage [126] and chemotherapy-induced cognitive decline [62]. Other reported risk factors for cognitive impairment following chemotherapy include younger age and multiple courses of chemotherapy [109,124]. While the effects of chemotherapy on cognitive outcomes in paediatric brain tumours need more targeted investigation involving cognitive testing at baseline, after any surgery and radiotherapy, the damage seen to WM can be identified radiologically.

### 5.9. Other Clinical and Demographic Risk Factors

Some risk factors cannot be predicted radiologically nor their long-term effects detected. Age at presentation [44,47,48,85,91], duration of symptoms [49,108], younger age when starting radiotherapy [75,108] and increasing time from radiotherapy [47,58,60] are proposed risk factors for cognitive impairment. The male sex has been associated with worse cognitive outcomes than females [61,85,89,108]. Population determinants of health, such as children from lower socioeconomic backgrounds [97], levels of parental education [44,97] and certain minority racial backgrounds within populations [127], have been associated with worse cognitive outcomes. Pre-existing clinical conditions associated with cognitive impairment must also be taken into consideration including developmental delay and autism spectrum disorder [128], but particularly those associated with brain tumour development, such as NF-1 [125] and tuberous sclerosis [129]. These risk factors highlight the benefits of integrating imaging variables with clinical and demographic patient data for cognitive outcome prediction.

## 6. How Has Imaging Contributed to Cognitive Outcome Prediction in Children?

While outcome prediction research is thriving, little has been carried out to produce clinical prediction models for cognitive outcomes in children with brain tumours particularly utilising brain imaging. The Neurological Predictor Scale (NPS) produces a composite score combining risk factors to predict cognitive and behavioural outcomes among paediatric brain tumour survivors [123]. It incorporates surgical interventions, radiotherapy, chemotherapy and secondary conditions including epilepsy, hormone deficiencies and hydrocephalus. It has demonstrated modest success in predicting IQ, processing speed, working memory, attention, abstract visual reasoning and academic measures [123,130,131,132]. However, it fails to assess risk factors relating to tumour variables and eloquent brain. That said, increasing NPS score has also been correlated with lower WM integrity based on reduced FA on DTI [131]. Integrating NPS score with tumour size improves prediction of cognitive outcomes [44], highlighting how radiological biomarkers can enhance cognitive outcome prediction. Decreased WM FA is frequently correlated with cognitive impairment. A meta-analysis investigating DTI indices and cognitive impairment among paediatric brain tumour and acute lymphoblastic leukaemia patients post-radiotherapy proposed WM integrity within the corpus callosum as a reliable biomarker for radiation-induced cognitive impairment [73]. Troudi et al. analysed perfusion using ASL techniques in 60 children, comparing infratentorial tumour patients with and without radiotherapy against healthy controls [25]. They reported that decreased hippocampal perfusion in their radiotherapy cohort was a radiological biomarker predictive of memory impairment.

pCMS is an important risk factor for cognitive impairment among infratentorial tumour patients. Spiteri et al. performed longitudinal semi-automated analysis of the inferior olivary nuclei on pre-, intra- and post-operative T2 MRI using a support vector machine combined with a generative k-nearest neighbour algorithm to select the most important variables from MRI features, radiological data and patient demographics [114]. Increasing hyperintensity in the left inferior olivary nucleus on serial imaging predicted pCMS six times more accurately than radiologists’ assessment. Spiteri et al. adopted the same machine learning techniques using fully automated image analysis of potential radiological pCMS biomarkers [133]. They identified seven possible biomarkers based on brain deformations and grey level changes within the brainstem and cerebellum following surgery using longitudinal T2 MRI. Liu et al. developed a predictive model for pCMS assessing imaging features, age and gender using a C4 Decision Tree Classifier in 89 patients with infratentorial tumours [113]. Using pre-operative MRI scans, they had 91% success in predicting pCMS based on radiologist assessment of cerebellar hemisphere invasion, middle cerebellar peduncle (MCP) invasion, dentate nuclei invasion, radiological diagnosis of ependymoma and MCP compression combined with age. These provide examples of how radiological analysis can be combined with machine learning to predict cognitive outcomes. Large-scale multi-site prospective longitudinal research is needed, focusing on the radiological, clinical and demographic risk factors, analysed using machine or deep learning techniques to produce clinical prediction models for cognitive impairment (Figure 8).

## 7. Challenges and Future Directions

### 7.1. Small Heterogenous Cohorts

Children with paediatric brain tumours form a heterogenous patient cohort with multiple potential confounding risk factors for cognitive impairment [8,19]. There are many tumour types and subtypes occurring in different locations at various stages of brain and educational development [134]. The analysis of so many competing variables can challenge conventional statistical methodologies [135]. Machine and deep learning techniques are valuable tools for complex research questions, but often require large patient numbers [37,45,135]. There is significantly less research focusing on paediatric neuro-oncology outcome prediction compared to adults, which likely relates to smaller paediatric patient numbers [136,137]. However, paediatric tumours differ from adult tumours [138] and occur during brain development, so insights from adult neuro-oncology are not directly transferrable to children. Consequently, most paediatric papers rely on small single-centre studies [8,9,73]. Artificial intelligence techniques can be used to analyse multi-dimensional small datasets [139] by internal resampling of training data [140] or the use of artificial augmentation techniques to increase the number of training images within limited samples [15]. However, using small single-centre datasets to train prediction models increases the risk of overfitting and reduces the generalisability to other populations [20,135,141]. Penalisation and shrinkage methods can counter this, but again are unreliable with very small datasets [141].

### 7.2. Difficulties Associated with Imaging in Children

Poor compliance with imaging and the need for sedation or general anaesthesia in children can limit the use of sequences with long acquisition times. While small, the risks to children requiring sedation or anaesthesia mean imaging is often limited to sequences that can be performed during scans for clinical reasons [134,142]. However, familiarisation strategies, noise reduction and paediatric-dedicated environments can reduce the need for sedation [142]. Clinical acquisition of DTI, perfusion and MRS sequences can be limited by long scanning and post-processing times. The desire for gold-standard imaging and research is balanced with other service delivery needs [143]. SIOPE guidelines [144] for paediatric brain tumour imaging recognise this, with different essential imaging guidance for hospitals with 1.5T and 3T MRI scanners due to the longer scanning times using 1.5T scanners. However, technological advances in MRI hardware are reducing scan times and increasing the feasibility of routine acquisition of advanced physiological imaging sequences. Recent artificial intelligence techniques involving denoising and image interpolation can also reduce scan times, while synthetic MRI enables multiple MRI tissue contrasts to be collected per acquisition [142]. Hospital expertise, scan and post-processing techniques vary [145]. Variable methodologies and research software not approved for clinical use can limit translation of research findings to clinical practice. Image-guidance software used for surgical and radiotherapy planning has been approved for clinical use and can perform automated segmentation of tracts (Figure 4), streamlining this process. Accuracy concerns can limit its use [145], but incremental advances are likely to increase its accuracy and viability.

### 7.3. Use of Historical Multi-Site Data

Investigating long-term outcomes with limited patient numbers often requires multi-site collaboration over extended time periods for patient enrolment and follow-up [135,144]. Consequently, organising and funding prospective research can be challenging. Instead, most research investigating long-term outcomes is retrospective [146]. However, using historical imaging datasets from multiple sites creates logistical challenges, which limits most research to single-centre studies or radiological assessment of tumour location. Advanced imaging sequences are often unavailable, and imaging acquired in different centres vary in terms of sequences performed, slice thickness, voxel size and planes of view [134]. The introduction of SIOPE [144] and RAPNO [147] guidelines attempts to standardise paediatric brain tumour imaging for clinical and research purposes. This should improve the comparability of imaging performed, benefitting future collaborative research, but many historical scans pre-date these guidelines. Another challenge is that different MRI scanners and local imaging protocols produce images that may look similar but have dramatically different signal intensity values within corresponding voxels [36,37]. Scans need harmonisation to standardise data, facilitating shared analysis, but there is no consensus for harmonisation techniques [21,135]. Consequently, many historical datasets are not suited to modern radiomic analysis or segmentation algorithms without significant post-acquisition processing. Cognitive outcome measures used also vary [8,19] with differing batteries of quantitative neuropsychological tests or qualitative outcome measures, including patient and carer surveys [148]. Guidelines for paediatric cognitive assessments produced for clinical trials will benefit international collaborative cognitive research [50,51,148], increasing patient numbers and generalisability of research findings [20]. While this has enabled prospective cognitive research, most studies in paediatric neuro-oncology still fail to utilise brain imaging [62,92].

Another hurdle is the need for data-sharing agreements between sites and ethical concerns regarding data privacy [9]. One option that bypasses the need to share sensitive patient data is federated learning techniques where the data analysis is performed locally and the results shared for combined analysis [9,20,35]. A solution to overcome differences in scanning parameters is for humans to analyse the scans and extract the pertinent data. With some training and a standardised reporting tool or lexicon, people can extract data reporting on the presence of features in scans, classifying the severity of variables or performing basic measurements such as length [35,36]. However, they need to be repeatable with high intra- and inter-observer variability [37]. Using language modelling, these data can be recorded within a database, integrated with clinical and demographic data, and formatted for advanced statistical, machine or deep learning analysis [45]. Within paediatric neuro-oncology, these techniques have been applied to predicting tumour type [149], classifying signs of raised intracranial pressure [49] and predicting pCMS prior to surgery [113]. The development of registries provides a valuable platform for data collection to be performed locally and then shared for central analysis. This has been applied to CT findings in traumatic brain injury without transferring or harmonising scans [150]. These techniques may increase collaboration and allow more rapid clinical translation of outcome prediction models produced in the current healthcare environment.

### 7.4. Gaps in the Research: Why These Matter and Solutions

At present, cognitive imaging research in paediatric neuro-oncology is limited in regard to identifying potential radiological biomarkers associated with cognitive outcomes. Multiparametric brain imaging has been used to predict the grade [11,12], type [12,13,15,22] and subtype of tumours [16,38,39,40,41,42] as well as prognosis [17,22] and recurrence risk [18,22], but it is not specifically targeted to cognitive outcomes per se. Clinical prediction models focusing on cognitive outcome prediction have not included radiological biomarkers [123] or involve only basic assessments like maximal tumour diameter [44] and presence or absence of hydrocephalus [53,89]. Research integrating different anatomical and physiological radiological features, combining them with clinical and demographic risk factors, is required. The machine learning and hence artificial intelligence techniques capable of producing these clinical prediction models will require data from large multi-institution collaborative studies to adequately power such research-driven models [45,135]. Hence, the recent introduction of guidelines for standardised imaging [144,147] and cognitive test measures [50,51] is likely to benefit future collaborative research. However, the variability in imaging performed in existing historical datasets requires advanced harmonisation techniques. An alternative approach bypassing the need for image harmonisation and sharing of sensitive patient data is for humans to analyse and in effect “code” imaging data in a reproducible and standardised way that does not detract from normal workflow patterns. The outcome would be to produce clinical prediction tools that can guide clinical decision-making for paediatric neuro-oncology multidisciplinary teams, both in terms of minimising the risk of cognitive impairment or permitting early identification of at-risk patients and hence the clinical provision of intensive support to aid this at-risk cohort. However, it is the future promise of the integration of such scoring and prediction methodological systems into radiological PACS systems and electronic medical record platforms that holds the most promise for maximising the neurocognitive outcomes for these children. From each patient’s imaging and medical records, it should also be possible to predict risk of tumour recurrence, endocrine abnormalities, physical disability and neurocognitive impairment in the form of risk stratification if we are to advance this field of personalised healthcare.

## 8. Conclusions

Multiparametric MRI scans contain vast amounts of data relating to tumour and brain physiology, which can be used to investigate cognitive processes occurring in paediatric neuro-oncology. While the causes of cognitive impairment in children with brain tumours are complex, there are techniques capable of integrating relevant radiomic features with clinical and demographic data for machine or deep learning analysis. These have already been utilised to differentiate tumours and predict other clinical outcomes. Within paediatric neuro-oncology, there is a need for large collaborative and prospective longitudinal studies to drive this research. However, solutions exist for the challenges of image standardisation, cognitive testing protocols and image harmonisation to enable this. Our understanding of the pathophysiology underlying cognitive impairment is improving. We need adequately powered research studies capable of utilising radiomics and artificial intelligence techniques to produce more complete prediction models for cognitive outcomes in children with brain tumours. Through better understanding of the processes underlying cognition in paediatric brain tumour patients, treatments can be adapted to minimise cognitive impairment and early interventions provided to enhance recovery and improve children’s future quality of life.

## Figures and Tables

**Figure 1 cancers-17-00947-f001:**
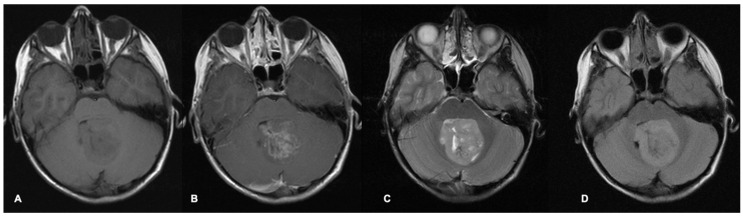
Axial MRI brain scan of a fourth-ventricle medulloblastoma in a 9-year-old boy. The pre-contrast T1 sequence (**A**) shows a heterogenous hypointense tumour filling and expanding the fourth ventricle, which demonstrates patchy contrast enhancement on the post-contrast T1 (**B**). The tumour appears mostly hyperintense with some heterogeneity on the T2 (**C**) and FLAIR sequence (**D**). FLAIR = fluid-attenuated inversion recovery.

**Figure 2 cancers-17-00947-f002:**
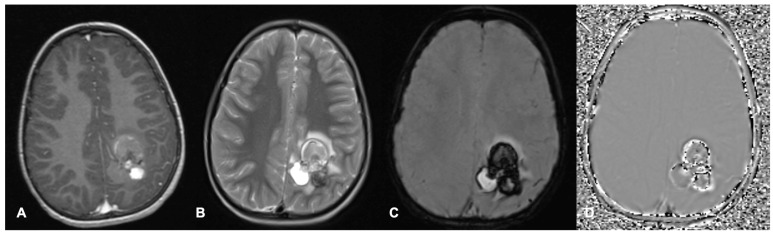
Axial MRI brain scan of a left parietal cavernous malformation in a 7-year-old boy. (**A**) shows a heterogeneously contrast-enhanced lesion on post-contrast T1. (**B**) Heterogenous T2 signal within the lesion with surrounding oedema. The lesion demonstrates increased susceptibility on SWI (**C**) with signal dropout in the filtered phase (**D**) in keeping with haemorrhage.

**Figure 3 cancers-17-00947-f003:**
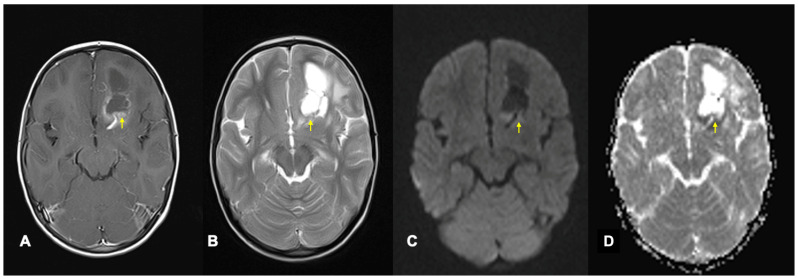
Axial MRI brain scan of a left frontal CNS neuroblastoma in a 4-year-old girl demonstrating diffusion restriction. (**A**) shows the post-contrast T1 sequence of a left frontal cystic lesion with an enhancing solid portion posteriorly. (**B**) is the T2 sequence with relative hyperintensity in the solid portion of the tumour. The solid portion appears hyperintense on DWI (**C**) and hypointense on ADC (**D**), in keeping with diffusion restriction. Areas of abnormality discussed are highlighted by yellow arrows. ADC = apparent diffusion coefficient; DWI = diffusion weighted image.

**Figure 4 cancers-17-00947-f004:**
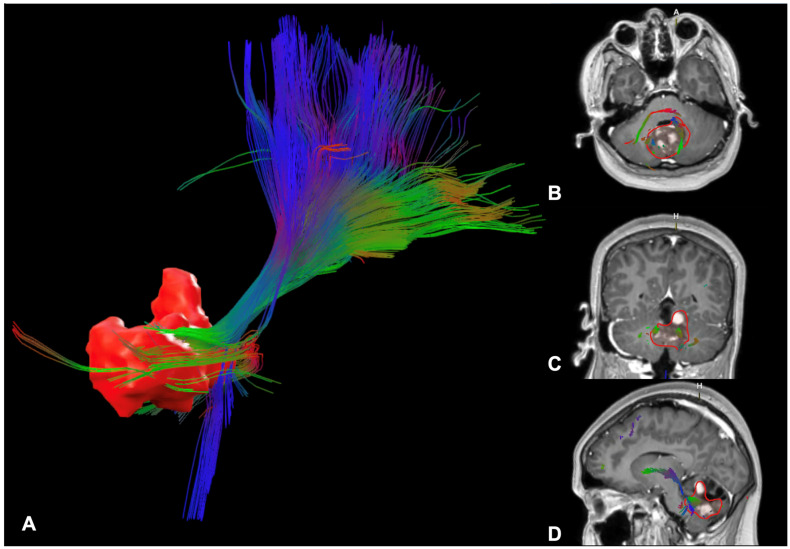
Tractography demonstrating WM fibres disrupted by a fourth-ventricular tumour in a 15-year-old female constructed from DTI using the Brainlab Elements^TM^ surgical planning software. (**A**) Three-dimensional reconstruction of the tumour (red) and the WM fibres including those involved in the dentato-thalamo-cortical tract viewed from a right lateral view. The right-hand column shows WM fibres and the tumour (outlined in red) laid over the post-contrast T1 sequence in the axial (**B**), coronal (**C**) and sagittal (**D**) planes. Orientation for the post-contrast sequences is indicated by A for Anterior and H for Head implying cranial.

**Figure 5 cancers-17-00947-f005:**
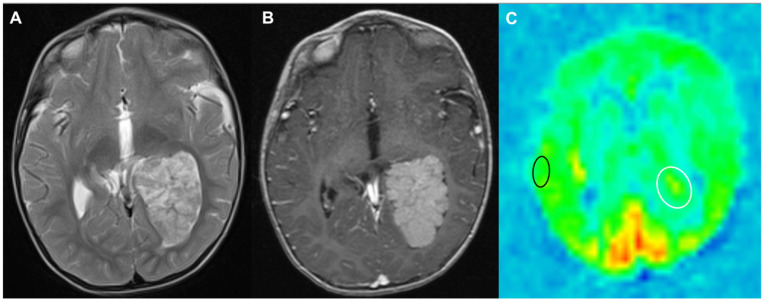
Axial T2-weighted image showing a mass lesion in the occipital horn of the left lateral ventricle in a 2-year-old boy (**A**). On the post-contrast T1-weighted image (**B**), the lesion has a lobulated margin with well-circumscribed contrast enhancement suggestive of a Choroid Plexus Papilloma. The ASL CBF image (**C**) shows that there is slightly increased perfusion (area within the white oval) within the tumour compared to the contra-lateral grey matter (area within the black oval). This is atypical, although perfusion is not as high as would be expected in a choroid plexus carcinoma. Pathological examination of the tumour indicated this was indeed an Atypical Choroid Plexus Papilloma with an elevated Chi 67 score.

**Figure 6 cancers-17-00947-f006:**
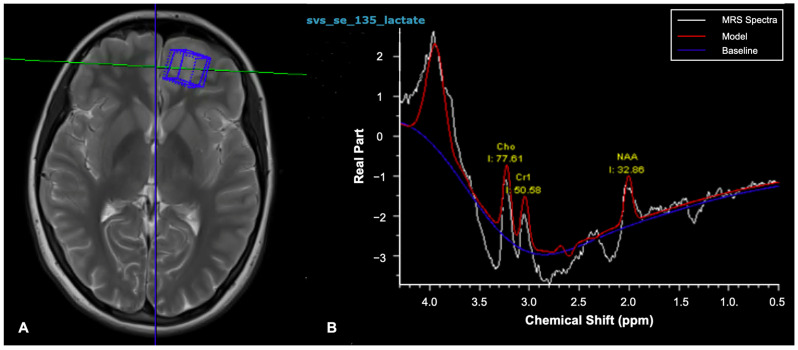
(**A**) Axial T2 MRI scan with a hyperintense left frontal ganglioglioma in a 15-year-old female demonstrating an ROI voxel for ^1^H MRS analysis. (**B**) ^1^H MRS analysis spectral graph based on the concentration of different chemicals within the ROI; the figure demonstrates the ^1^H spectra taken from a voxel within the left frontal tumour (TE = 135 ms). Ratios of metabolite concentrations are given in reference to lactate. Cho = choline-containing metabolites; Cr = creatine; NAA = N-acetyl aspartate; ppm = parts per million.

**Figure 7 cancers-17-00947-f007:**
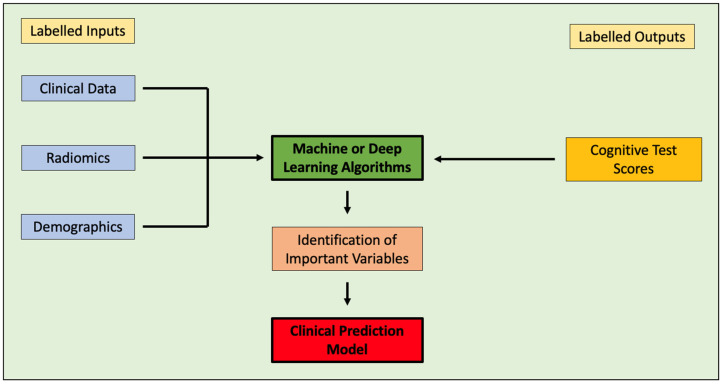
Flow diagram outlining how clinical, radiomic and demographic data can be integrated and analysed in patients with known cognitive test scores from a training dataset to identify risk factors and produce clinical prediction models using supervised machine and deep learning techniques. The performance of the clinical prediction model can then be analysed in a validation dataset.

**Figure 8 cancers-17-00947-f008:**
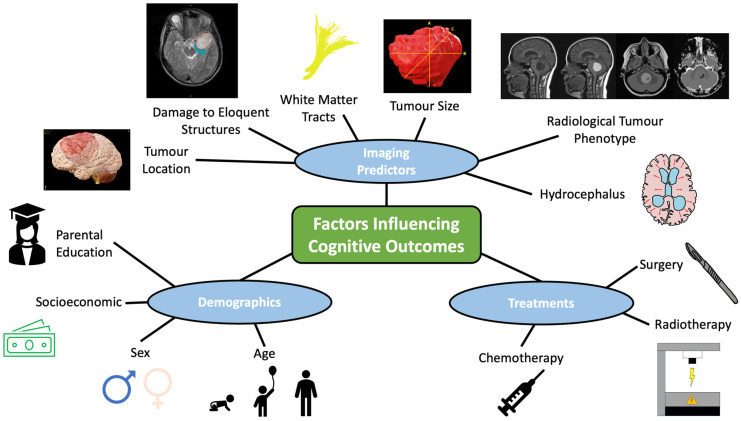
Spider diagram outlining some of the imaging, treatment, and demographic variables, which might contribute to clinical prediction models seeking to predict long-term cognitive outcomes in children with brain tumours.

**Table 1 cancers-17-00947-t001:** Common conventional MRI sequences and their applications in neuro-oncology. These can display tumour location and basic characteristics about the tumour, which can help differentiate between some tumour types.

MRI Sequence	Applications
T1	Demonstrates brain anatomy Pre-contrast T1 hyperintensity can be caused by fat, blood components and melanin Post-contrast enhancement demonstrates breakdown of the blood–brain barrier [10,23]
T2/FLAIR	Demonstrates brain anatomy T2 hyperintensity can be caused by oedema (infiltrative and vasogenic), WM injury, non-enhancing tumours and gliosis [10,23] FLAIR can visualise periventricular WM lesions by suppressing T2 hyperintensity in free water including CSF
T2*/SWI	Demonstrates magnetic susceptibility Susceptibility can be produced by calcification and blood products from haemorrhage or increased vascularity [10,23] Blood is paramagnetic so appears hypointense on filtered phase imaging, distinguishing it from calcium, which is diamagnetic and appears hyperintense [10]

CSF = cerebrospinal fluid; FLAIR = fluid-attenuated inversion recovery; SWI = susceptibility-weighted imaging; WM = white matter.

**Table 2 cancers-17-00947-t002:** Advanced MRI sequences and their applications in neuro-oncology. These sequences can provide information about the physiology of the tumour. In combination with the conventional MR sequences, they can help differentiate between many different types of brain tumours and demonstrate physiological changes occurring within the brain.

MRI Sequence	Applications
**Diffusion-Weighted** **Imaging**	Quantifies diffusion of water molecules within tissue Restricted diffusion can be caused by high tumour cellularity, acute surgical damage and cytotoxic oedema [10,24]
**Diffusion Tensor Imaging**	Displays main diffusion direction and thus WM fibres Can quantify WM integrity at presentation and post-treatment Tractography can locate WM tracts for surgical planning, identify shifts and tumour infiltration [10,24]
**Perfusion**	Can measure CBF, CBV and vascular permeability Increased perfusion can be a marker of high-grade tumours, areas of transformation and recurrence [10,24] Reduced CBV in brain structures can be a marker of radiation-induced chronic small-vessel disease [25]
**Magnetic Resonance Spectroscopy**	Analyses tumour biochemistry indicating tumour metabolism and physiology Increased Cho and decreased NAA levels are seen in tumour cells with higher Cho/NAA and Cho/Cr ratios seen in higher-grade tumours [10,26] Lactate and lipid peaks represent necrosis and hypoxia seen in high-grade tumours [10,26]

CBF = cerebral blood flow; Cr = creatinine; CBV = cerebral blood volume; Cho = choline, NAA = N-acetylaspartate; WM = white matter.

**Table 3 cancers-17-00947-t003:** Different applications of multiparametric MRI in paediatric neuro-oncology research and examples.

Applications	Examples
**Tumour Grade**	High-grade vs. low-grade [11,12]
**Tumour Type**	Medulloblastoma vs. ependymoma vs. pilocytic astrocytoma [12,13] Medulloblastoma vs. pineoblastoma vs. supratentorial primitive neuroectoderm tumours [22] Adamantinomatous craniopharyngioma vs. other sella and suprasellar tumours [15]
**Radiogenotyping/** **Molecular Subtyping**	Medulloblastoma [16] Ependymoma [38] Atypical Teratoid Rhabdoid Tumour (ATRT) [39] Diffuse Intrinsic Pontine Glioma (DIPG) [40] Paediatric Low-Grade Glioma (pLGG) [41] Choroid Plexus Papilloma [42]
**Prognostication**	Risk of neuroaxis metastasis [22] Risk of recurrence [18,22] Survival [17]

**Table 4 cancers-17-00947-t004:** Overview of existing brain eloquence classifications in neuro-oncology and neurosurgery.

Name	Population	Grading System	Classification
**Sawaya** [71]	Adult HGG	Graded Eloquence— 1–3	Grade 1 (non-eloquent)—frontal or temporal pole, right fronto-parietal, cerebellar hemisphere Grade 2 (near-eloquent)—near motor or sensory cortex, near calcarine fissure, corpus callosum, near brainstem Grade 3 (eloquent)—motor or sensory cortex, speech centre, basal ganglia, internal capsule, hypothalamus/thalamus, brainstem and dentate nucleus
**Friedlein** [72]	Adult HGG	Resectability—A/B	Grade A—Sawaya 1 and 2—resectable Grade B—Sawaya 3—unresectable
**Shinoda** [70]	Adult Supratentorial HGG	Group—A/B/C and Eloquent—Y/N	Group A—Left occipital and right-sided cortical tumours (to the depth of the globus pallidus) Group B—Left frontal, parietal and temporal tumours (to the depth of the globus pallidus) Group C—Midline tumours and those involving the other deep grey nuclei and internal capsule Eloquent structures being those adjacent to the sensorimotor and language cortex, corpus callosum, thalamus, internal capsule, fornix, hypothalamus and brainstem
**Chang** [69]	Adult Hemispheric LGG	Eloquent—Y/N	Eloquent brain being the pre- and post-central gyri, perisylvian language cortex of the dominant hemisphere, thalamus, internal capsule, basal ganglia and calcarine cortex
**Spetzler-** **Martin** [68]	Arteriovenous Malformations	Eloquent—Y/N	Eloquent brain being primary sensorimotor cortex, visual cortex, language cortex, internal capsule, thalamus, hypothalamus, brainstem, cerebellar peduncles and cerebellar nuclei

Y/N = yes/no.

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
