# Peer review of "Radiological Predictors of Cognitive Impairment in Paediatric Brain Tumours Using Multiparametric Magnetic Resonance Imaging: A Review of Current Practice, Challenges and Future Directions"

_cancers, 2025, doi:10.3390/cancers17060947_

Round 1

Reviewer 1 Report

Comments and Suggestions for Authors

Summary

The paper is well written and in my opinion very interesting. This article shows the techniques of differential diagnosis (MRI) and treatment of brain tumors in childhood.

Intringuily authors showed how cognitive damage is predicted by means of deep learning and algorithms, based on demographic and clinical data. They explained how much MRI imaging influences outcomes togheter with large size sample.

Finally they explained how to improve the quality of the incoming MRI data: by increasing the sample size, adding the functional sequences DTI, MRS, perfusion and introducing standard protocols for multicenter studies.

As follows I have shown sections that need to be improved in my opinion.

2.2. Advanced MRI

I kindly ask the authors to provide bibliographic references for the Magnetic Resonance Spectroscopy point 2 in table 2 (pag 4).

I suggest to discuss the considerable influence of the axial diffusivity (AD) to identify white matter fasciculus in DTI  paragraph (pag 5) and ouline its relationship with radial diffusivity (RD).

The operator influence of ROI selection to identify nerve fiber bundles should be discussed too in probabilistic tractography method.

Figure 4 should show more clear fiber tracts, I answer to replace it.

“The height of generated peaks…” (192, pag 5) have be replaced with the “area under generated peaks…”, in agreement with fig. 6.

Author Response

Comment 1 : I kindly ask the authors to provide bibliographic references for the Magnetic Resonance Spectroscopy point 2 in table 2 (pag 4).

Response: We agree. This has been added.

Comment 2: I suggest to discuss the considerable influence of the axial diffusivity (AD) to identify white matter fasciculus in DTI paragraph (pag 5) and ouline its relationship with radial diffusivity (RD).

Response: We agree. We discuss the influence of the axial diffusivity (AD) and radial diffusivity (RD), which we have now outlined (pag 5).  'Diffusion parallel to WM fibres can be measured with axial diffusivity (AD), while diffusion perpendicular to WM fibres can be measured with radial diffusivity (RD) [29].'

Comment 3: The operator influence of ROI selection to identify nerve fiber bundles should be discussed too in probabilistic tractography method.

Response: We agree. We discuss that operator influence of ROI selection for identification of WM fibres when using probabilistic tractography methods. We have added ". More advanced probabilistic models can account for crossing fibres [24], but they are user-dependent as they are traced from a selected starting point [30]." Line 160-161.

Comment 4: Figure 4 should show more clear fiber tracts, I answer to replace it.

Response: We agree. We have replaced this figure with a new one.

Comment 5: “The height of generated peaks…” (192, pag 5) have be replaced with the “area under generated peaks…”, in agreement with fig. 6.

Response: We agree. This has been updated.

Reviewer 2 Report

Comments and Suggestions for Authors

The imaging and accurate diagnosis of pediatric brain tumors present significant radiological challenges, with magnetic resonance imaging (MRI) playing a pivotal role in providing tumor-specific imaging information. MRI is routinely performed for children suspected of having a brain tumor at the time of initial presentation. In this paper, the authors review how brain imaging can be utilized to predict cognitive impairment and discuss the challenges and potential solutions in this area of research. These challenges include the need for large patient cohorts, which necessitates multi-site collaboration, as well as variations in imaging protocols.

Comments:

  1. The authors referenced over 250 publications in this manuscript.
  2. Separate tabulations for each category could be included to enhance clarity and organization.
  3. The research gap identified and the proposed solutions should be explicitly outlined to strengthen the manuscript.
Comments on the Quality of English Language

Minor editing is required

Author Response

Comment 1: The authors referenced over 250 publications in this manuscript.

Response: We agree. We have reduced the number of references from 255 to 150.

Comment 2: Separate tabulations for each category could be included to enhance clarity and organization.

Response: We agree and have added 2 tables. “3. Applications of Advanced Imaging Analysis for Prediction in Paediatric Neuro-oncology”  on page 9. titled “Table 3. Different applications of multiparametric MRI in paediatric neuro-oncology research and examples.” The second table is titled on page 11 “Table 4. Overview of existing brain eloquence classifications in Neuro-oncology and Neurosurgery.” within 4.2. Eloquence Classifications”.

Comment 3: The research gap identified and the proposed solutions should be explicitly outlined to strengthen the manuscript.

Response: We agree and have added a section  “6.4 Gaps in the research, why this matters and solutions” on page 19.

Round 2

Reviewer 2 Report

Comments and Suggestions for Authors

Suggestions mentioned in the previous review

  1. The authors referenced over 250 publications in this manuscript.
  2. Separate tabulations for each category could be included to enhance clarity and organization.
  3. The research gap identified, and the proposed solutions should be explicitly outlined to strengthen the manuscript.

The authors incorporated all the suggestions mentioned in the previous review.

Comments on the Quality of English Language

The English could be improved to more clearly express the research.

Author Response

Thank you for your further feedback. As per editor Ms Lin, we have added a material and methods section. We have re-read the document in its entirety improving grammar and spelling. If you feel any other changes are needed in reference to the English, please let us know what changes you feel are necessary,